# Prevalence of Non-Viral Bloodborne Pathogens Among Healthy Blood Donors in Western Mexico: Problems and Failures of Public Health Policy

**DOI:** 10.3390/pathogens13121027

**Published:** 2024-11-21

**Authors:** José de Jesús Guerrero-García, Alejandra Flores-González, Alma Marina Sánchez-Sánchez, Rafael Magaña-Duarte, Mario Alberto Mireles-Ramírez, Pablo Cesar Ortiz-Lazareno, Erick Sierra Díaz, Daniel Ortuño-Sahagún

**Affiliations:** 1Banco de Sangre Central, Unidad Médica de Alta Especialidad (UMAE), Hospital de Especialidades (HE), Centro Médico Nacional de Occidente (CMNO), Instituto Mexicano del Seguro Social (IMSS), Guadalajara 44340, Mexico; jose.guerrero9683@academicos.udg.mx (J.d.J.G.-G.); rafael.magana@imss.gob.mx (R.M.-D.); 2Departamento de Farmacobiología, Centro Universitario de Ciencias Exactas e Ingenierías (CUCEI), Universidad de Guadalajara, Guadalajara 44340, Mexico; alejandra.flores7408@alumnos.udg.mx; 3Laboratorio de Neuroinmunobiología Molecular, Instituto de Investigación en Ciencias Biomédicas (IICB), Centro Universitario de Ciencias de la Salud (CUCS), Universidad de Guadalajara, Guadalajara 44340, Mexico; marina.sanchez@academicos.udg.mx; 4Dirección de Investigación y Educación en Salud, Unidad Médica de Alta Especialidad (UMAE), Hospital de Especialidades (HE), Centro Médico Nacional de Occidente (CMNO), Instituto Mexicano del Seguro Social (IMSS), Guadalajara 44340, Mexico; mario.mirelesr@imss.gob.mx; 5División de Inmunología, Centro de Investigación Biomédica de Occidente (CIBO), Instituto Mexicano del Seguro Social (IMSS), Guadalajara 44340, Mexico; pablo.ortiz@imss.gob.mx; 6División de Epidemiología, Unidad Médica de Alta Especialidad (UMAE), Hospital de Especialidades (HE), Centro Médico Nacional de Occidente (CMNO), Instituto Mexicano del Seguro Social (IMSS), Guadalajara 44340, Mexico

**Keywords:** bloodborne pathogens, syphilis, Chagas disease, brucellosis, blood donors, Mexico

## Abstract

Background and Objectives: Non-viral bloodborne diseases are a group of infections that are a public health problem worldwide. The incidence of diseases such as brucellosis and syphilis is increasing in the Americas and Europe. Chagas disease is an endemic problem in Latin America, the United States and Europe. This study aims to determine the prevalence of non-viral bloodborne diseases in blood donors and to discuss some issues related to federal regulations for the control and prevention of these infectious diseases in Mexico. Material and methods: A cross-sectional study was conducted in the Western National Medical Center Blood Bank, including 228,328 blood donors (2018–2023). Frequencies, percentages, means, standard deviation and confidence intervals (CI) were calculated for demographic data. Prevalences were expressed as rates per 100,000 with 95% CI. Results: Of 3949 seroreactive or undetermined blood donors at the first screening, a total of 682 (0.299%) completed their follow-up test and were positive for *Treponema pallidum* (478), *Trypanosoma cruzi* (83), or *Brucella* spp. (121). The overall prevalence for non-viral bloodborne diseases was 299 per 100,000 blood donors. The prevalence for syphilis, Chagas disease, and *Brucella* was 209, 36, and 53 per 100,000 respectively. Conclusion: Federal regulations should be reviewed to formulate specific public health policies focused on controlling and preventing nonviral bloodborne diseases.

## 1. Introduction

One of the most important goals of a blood bank is to provide safe blood, as bloodborne pathogens transmitted by transfusion are a serious problem in less developed and developing countries, where the financial resources allocated to healthcare are limited. In addition, not all of these countries have a fully developed policy or legal framework for the safety and quality of blood transfusions. To reduce or mitigate the risk, the World Health Organization (WHO) recommends testing all blood donations for bloodborne pathogens, including human immunodeficiency virus (HIV), hepatitis B virus (HBV), and hepatitis C virus (HCV) and *Treponema pallidum* (syphilis) [1]. We previously reported on the seroprevalences of HIV, HCV, and HBV in the blood donor population [2]. Furthermore, most prevalence studies focus on viral pathogens, and often overlook others that are endemic in middle- and low-income countries, such as Chagas and *Brucella*.

In addition, blood banks around the world can also test for region-specific pathogens, such as *Babesia microti*, an intra-erythrocytic parasite transmitted by ticks. Cases of *B. microti* have been reported in the northeastern and northern midwestern United States (Connecticut, Massachusetts, Maryland, Maine, Minnesota, New Hampshire, New Jersey, New York, Pennsylvania, Virginia, Vermont, Wisconsin and Washington DC) [3,4]. In Mexico, syphilis, Chagas disease and brucellosis pose serious transmission risks from contaminated blood transfusions, which have a negative impact on public health and lead to the spread of disease and severe and chronic complications in the recipient population. In addition, Mexico is considered an endemic country for Chagas disease due to its climate and the presence of triatomines [5], and for brucellosis due to the high consumption of unpasteurized dairy products. To ensure the safety of the blood supply and protect public health from these potentially serious infections, it is therefore essential to test blood donations for these diseases. This proactive approach helps to reduce the incidence of transfusion-transmitted infections and the associated health burden. In this sense, federal regulations (NOM-253-SSA1_2012) stipulate that all blood donors must be tested for HIV, HBV, HCV, syphilis, *Trypanosoma cruzi* (Chagas disease) and—depending on the region and epidemiology—for other bloodborne pathogens: *Brucella*, *Plasmodium*, Cytomegalovirus, *Toxoplasma* and Human T-lymphotropic virus (HTLV) 1 and 2. The NOM-253-SSA1-2012 is the national framework that governs the blood supply and emphasizes the importance of encouraging voluntary, unpaid blood donations to ensure a safe and sufficient supply. It establishes strict standards for the collection, testing, processing and distribution of blood and its components to minimize transfusion-transmitted infections [6].

This screening is critical for the safety and availability of blood and for the provision of high-quality blood services for transfusions. As in other countries around the world, the legal framework in Mexico requires that a second sample be collected from reactive or undetermined donors for re-screening and confirmatory test. This follow-up can prevent the future collection of contaminated units, and allows the national health system to detect cases of infectious diseases in the population in a timely manner. In addition, donor testing in blood banks provides useful insights into epidemiology of infectious diseases in the general population worldwide and in the context of specific countries, particularly for non-viral bloodborne pathogens. These data enable the government to develop new public health measures or update existing ones. It should be noted that Chagas disease has become increasingly important in the United States due to the migration of people from other countries in the Americas [5].

The Western National Medical Center in Guadalajara provides health services for approximately 17,000,000 beneficiaries of the Mexican Social Security Institute (IMSS). The medical center Central Blood Bank is the second largest blood bank in Mexico. In 2020 and 2021, it was the national leader in blood collection and the production of blood components in the country. In the last five years, the Central Blood Bank has collected more than a quarter of a million (284,694) donations from 228,328 donors, including first-time and repeat donors. These numbers have made the Central Blood Bank a national reference point for the epidemiology of bloodborne diseases. The aim of this study was to determine the prevalence of three major non-viral bloodborne diseases among blood donors and, on this basis, to discuss some issues related to public health control and prevention strategies.

## 2. Materials and Methods

A cross-sectional study was conducted using the Western National Medical Center Blood Bank database with institutional approval (R-2023-1301-074/COFEPRIS 17 CI 14 039 114). The database included blood donors from August 2018 to July 2023. All blood donors were between 18 and 65 years of age, interviewed by physicians, and screened for viral and non-viral bloodborne pathogens (HIV, HCV, HBC, *Trypanosoma cruzi*, *Treponema pallidum* and *Brucella*), according to national and institutional regulations [6]. Positive results for viral diseases were not considered for this study.

Serological screening for anti-*Treponema pallidum* and anti-*Trypanosoma cruzi* antibodies was performed as advised by the manufacturer, using the automated microparticle chemiluminescence immunoassay (CIA) platform ABBOTT Architect i4000 with Architect System Syphilis TP (specificity 99.94%, CI 95% 99.83–99.9%; sensitivity ≥ 99.9%) and Chagas (specificity 99.96%, CI 95% 99.90–99.99%; sensitivity 99.74%) assays kits. Samples were classified as “reactive” when S/CO values were ≥1.00 for syphilis and Chagas, and as “undetermined” when S/CO values were 0.90 to 0.99, and 0.80 to 0.99, respectively. For brucellosis screening, the LICON Bengal Rose test was performed, a rapid agglutination test using stained *Brucella abortus* biotype 1 (strain 995 or strain 119-3) suspension (specificity 98.5%, sensitivity 99%). If a sample was reactive or undetermined for *Treponema pallidum* or *Trypanosoma cruzi*, a sample of the plasma was collected as prescribed by the manufacturer, and the analysis was repeated for both samples simultaneously. For reactive *Brucella* samples, the analysis was repeated only in the donor sample, because the Bengal rose test cannot be performed in anticoagulated samples.

Regarding NOM-253-SSA1-2012, donor follow-up was undertaken when a reactive or undetermined result was identified at initial screening. At least three attempts were made to locate the donor to obtain a second sample within 8 days, and screening was repeated. If reactivity was confirmed in this second sample, it was further analyzed with the appropriate confirmatory test. The confirmatory tests were (1) detection of anti-*Treponema pallidum* by passive hemagglutination BIO-RAD TPHA (specificity 99.72%, CI 95% 99.53–99.85%; sensitivity 100%, CI 95% 99.2–100.0%) [7,8], (2) detection of anti-*Trypanosoma cruzi* hemagglutination Wiener lab Chagatest HAI (specificity and sensitivity 95%), and (3) 2-mercaptoethanol *Brucella* agglutination test [6]. Donors with positive confirmatory tests were considered positive cases of infection, and were referred to the epidemiology service of their medical care unit for clinical care and treatment.

Statistical analysis was performed using percentages, absolute frequencies, and measures of central location, as the means. The chi-square test was used to analyze the categorical data. We considered statistical significance and differences in means were reached when *p* < 0.05. All data were processed using Excel^®^ (Microsoft, Redmond, WA, USA) and Open Epi (Open-Source Epidemiologic Statistics for Public Health, Bill and Melinda Gates Foundation, Emory College, Atlanta, GA, USA). Prevalences were calculated based on confirmed positive cases (Figure 1) and were calculated as rates per 100,000 with 95% CI. For demographic data, frequencies, percentages, means, standard deviation and confidence intervals (CI) were calculated. For the calculation of prevalence, only those donors who had completed their follow-up from the initial screening at the time of donation until the confirmatory test result was available were included.

## 3. Results

A total of 284,694 donations were collected from 228,328 donors (186,509 first-time and 41,819 repeat donors). Of these donations, 228,328 were first-time donations (assuming a donor makes at least one first-time donation) and 56,366 were repeat donations. The mean age of the cohort was 33.84 years (SD 10.66, CI 95% 33.80–33.88). Of these, 146,866 donors (64.32%) were men (mean age 33.93, SD 10.60, CI 95% 33.88–33.98) and 81,462 (35.68%) were female (mean age 33.69, SD 10.77, CI 95% 33.62–33.76) (ratio 2:1). The sex distribution of first-time and repeat donors is shown in Table 1.

All blood donors were screened at the time of donation. We found 3643 seroreactive or undetermined donors for nonviral bloodborne pathogens (2276 for syphilis, 597 for Chagas disease, and 770 for *Brucella*). According to national regulations, reactive or undetermined donors must be notified so that a second sample can be collected, and the appropriate confirmatory test performed. However, only 1253 donors (34.4%) were located and their follow-up completed (Figure 1).

We confirmed 682 positive cases (0.299%) of non-viral bloodborne diseases (478 for syphilis, 83 for Chagas disease, and 121 for brucellosis) among the 228,328 donors. The prevalence for the three infections was 299 per 100,000 blood donors (CI 95% 277–322). The prevalence for the three infections in blood donors divided into males and females was 188 (CI 95% 171–206) and 111 (CI 95% 98–125) per 100,000, respectively. However, when we disaggregated blood donors by sex, the prevalence for the three infections was 293 (CI 95% 267–322) and 311 (CI 95% 276–352) per 100,000 for males and females, respectively (OR 0.9; CI 95% 0.81–1.10; *p* > 0.05). The prevalence for each nonviral bloodborne disease was 209 (CI 95% 191–229), 36 (CI 95% 29–45), and 53 (CI 95% 44–64) per 100,000 for syphilis, Chagas disease, and brucellosis, respectively. The analysis of prevalence for each non-viral bloodborne disease and sex-specific prevalence is shown in Table 2. Remarkably, only the prevalence of brucellosis depended on sex (OR 0.6; CI 95% 0.41–0.83; *p* < 0.01).

Finally, the data of positive cases and prevalence for first-time and repeat donors are presented in Table 3 and Figure 2. These data show that the prevalence for the three nonviral bloodborne diseases is higher in first-time (328 per 100,000; CI 95% 303–355) than in repeat donors (167 per 100,000, CI 95% 132–211). However, when prevalence was classified according to nonviral bloodborne diseases, the prevalence for syphilis and Chagas disease was higher in first-time donors. Conversely, the prevalence for brucellosis was higher in repeat donors (Table 3).

## 4. Discussion

Bloodborne diseases are a serious public health problem that threatens the availability and supply of safe blood in countries that lack the financial resources to implement the necessary blood testing and control measures. To reduce the risk of infection with bloodborne pathogens, the WHO recommends testing all blood donations for HIV, HCV, HBV, and syphilis [1]. This strategy may be useful as an epidemiologic tool in an attempt to detect positive cases in the asymptomatic population and provide timely care and treatment to this population. In addition to these four bloodborne diseases, in Mexico, it is mandatory to carry out screening for *Trypanosoma cruzi* and *Brucella* in those regions that are considered endemic [6]. In Mexico, tropical weather conditions, altitude, type of vegetation and land use influence the distribution of triatomines, insects that act as vectors in the transmission of Chagas disease, as it has been shown that temperature and rain can alter their behavior [9]. In addition, the high consumption of unpasteurized dairy products makes western Mexico an endemic area for brucellosis.

In this study, we present the results of the prevalence of syphilis, Chagas disease, and brucellosis obtained from screening and the confirmation results among blood donors in the Central Blood Bank of the Western National Medical Center of IMSS. This represents the profile of the western population of Mexico, as the Central Blood Bank is the second most productive and largest blood bank in the country. Our analysis can be used to develop new and better health policies for these three bloodborne pathogens and transfusion medicine. Perhaps the numbers and prevalence values do not seem to show alarming results, but it is important to emphasize that even if the numbers and prevalence are not high, they comes from blood donors who described themselves as healthy and asymptomatic in an initial survey. Therefore, it can be assumed that there is a larger problem in the wider population, which could be a potential public health concern.

Syphilis belongs to the group of sexually transmitted diseases. Prevalence in the world may vary by region. In Sweden, the prevalence was reported to be 4.6 per 100,000 in 2021 [10]. The prevalence of syphilis is increasing in Europe and the United States. Diagnosis is difficult because the incubation period is 10 to 90 days and asymptomatic carriers are capable of transmission for a year. Several data have been obtained from screening blood donors [11]. The prevalence of syphilis among blood donors in Mexico varies, with some studies reporting rates of around 2.1% [12]. Screening for syphilis in blood donors has reduced the transmission of this pathogen through transfusions. However, there is always a residual risk for all bloodborne pathogens, even if *T. pallidum* is susceptible to cold storage. In this sense, it has been demonstrated that the infectivity of *T. pallidum* is maintained in blood components for several days [13,14]. However, it is important to keep in mind that the detection of anti-*T. pallidum* antibodies by CIA has a higher false-positive rate in populations with low syphilis prevalence [15]. In addition, it is worth noting that non-venereal treponematoses such as soft palate, endemic syphilis and pinta can cause false-positive results in syphilis screening and confirmation tests [16].

The World Health Organization’s Global Health Sector Strategy has identified the problem of syphilis as a target to reduce incidence by 90% by 2030 [17,18]. A meta-analysis identified various prevalence rates worldwide between 2000 and 2020 among men who have sex with men (MSM), a population at higher risk of any sexual transmitted infection than blood donors. In this study, global prevalence was estimated at 7.5%, with the lower rates reported in New Zealand and Australia (1.9%), Central and South Asia (5.0%), and the highest rates in Latin America and the Caribbean (10.6%). Prevalence rates between 5 and 10% were reported for Europe, Canada, the United States, and Mexico. Countries such as Argentina and Pakistan had prevalence rates greater than 20% [19]. In general, the prevalence of syphilis in our blood donors (209 per 100,000, 0.209%) is, as expected, lower than the prevalence reported by Tsuboi et al., who focused on the MSM population. The regulations for blood banks in Mexico take into account the exclusion of donors based on risk factors related to sexual practices, such as contact with secretions and body fluids of persons who may have sexually transmitted diseases. However, these results are underestimated by the existence of a bias, as not all reactive/unsafe blood donors complete their follow-up, and positive cases cannot be confirmed, as we explain later, when dealing with the limitations of the study. However, analysis of the data suggests that syphilis prevalence is not dependent on donor gender. It is therefore possible that donors who did not complete their follow-up may maintain this statistical trend.

Previous studies show that the prevalence of syphilis varies depending on the geographical region and income of countries. For example, Quintas et al. reported high syphilis rates in blood donors from Luanda, Angola (20%) [20] and Alharazi et al. reported high syphilis rates in Yemen (2.4%) [21] On the other hand, Niederhauser et al. conducted an analysis of 9,753,402 blood donations from Switzerland received between 1996 and 2001. Of these donations, 377 positive syphilis cases were confirmed in first-time donors and 241 in repeat donors. It should be noted that in this study, 92.5% of donations came from repeat donors and only 7.5% from first-time donors [22], in contrast to our population, where the majority of donations in Mexico come from first-time donors. According to the study by Osei-Boakye et.al. (2024), first-time donors are more likely to report high-risk behaviors such as multiple sexual partners, drug use, and recent tattoos or piercings. In addition, first-time donors have a higher deferral rate due to these behaviors and other factors such as low hemoglobin levels. Repeat donors, on the other hand, generally report less risky behaviors and are better informed about safe practices. In addition, repeat donors generally have a healthier profile, fewer refusals and better overall health indicators [23]. According to the report by Conti et al., the prevalence of syphilis among blood donors from four major blood donation organizations in the United States (the American Red Cross, New York Blood Center Enterprises, OneBlood and Vitalant) from 2020 to 2022 was 28.4 per 100,000 [24]. These data are consistent with the fact that transfusion-transmitted infections are more common in low- and middle-income countries, including Mexico (209 per 100,000, Table 2), than in high-income countries [25], indicating that Mexico has a long way to go to reach first-world levels of blood safety.

In 2023, the *Epidemiological Bulletin of Mexico* registered 161 cases of acquired syphilis [26]. However, the reported numbers in our country probably do not reflect the reality. Blood banks are obliged to report cases of blood-transmissible pathogens detected in donors. However, compliance with this obligation is limited by several factors, in particular the lack of follow-up of reactive donors. To avoid this, it is necessary to update national regulations to try to confirm a reactive blood donor from their first screening and to educate donors about the importance of their health status when they are called to be informed of a reactive result. In addition, physicians should be prepared to (1) recognize the disease and its risk factors when interviewing donor candidates and deciding whether or not they are eligible for donation, (2) continue to manage reactive donors and refer them immediately to their medical unit once the diagnosis is confirmed, and (3) reduce the incidence by providing information about the risks and responsible sexual practices.

Moreover, Chagas disease is an epidemiological surveillance infection and a public health problem in Mexico. The *Epidemiological Bulletin of Mexico* registered 990 new cases of Chagas disease (99 acute and 891 chronic) in 2023 [26], and these data seem far from the reality. The country is classified as an endemic area by the Pan American Health Organization (PAHO). The problem is not limited to Mexico and Latin America; the disease has also been detected in United States of America (USA), Canada, and Europe. In the United States alone, there are 240,000 to 350,000 estimated cases of Chagas disease, most of which are due to the phenomenon of migration of people from other countries in the Americas [5]. Although there are areas of the world where this disease is endemic due to climatic conditions, human migration and the ability of the parasite to use rodents and dogs as vectors [27] are factors that have influenced the spread of Chagas disease. Therefore, it has gained importance in the field of blood safety in Europe, where several countries have developed prevention strategies to avoid the transmission of Chagas through transfusion [28].

The infection is caused by *Trypanosoma cruzi*, a protozoan that is endemic to the Americas. Infected patients may develop chronic disease, and about 40% are characterized by cardiomyopathy and megacolon. Some patients may develop a stroke and polyneuropathy [29]. In 2012, the reported prevalence was 0.70 per 100,000, and a total of 150 cases were reported in the first 4 months in 2018. A study in Mexico involving 18,256,144 blood donors nationwide reported an average prevalence of 0.39% over 10 years (2007–2017) [30]. In blood donors, the prevalence of Chagas disease is estimated to be around 0.65% [12], but there are no studies that allow us to have an idea of the epidemiological situation in Mexico. The prevalence reported in the present study is far from the national numbers. The 83 cases reported in the last five years are not representative of the national level. However, the western region of Mexico has a lower prevalence of this disease than other states in the south of the country. The prevalence of Chagas disease in blood donors has also been reported in other Latin American countries. In Pará, Brazil, the reactive rate was 0.1%, but positive cases decreased to 0.07% [31]. In Colombia, a meta-analysis showed that the prevalence of Chagas disease varied according to clinical subgroups, with a pooled prevalence of 2.0%. Higher prevalences were observed in adults (3.0%) and pregnant women (3.0%), while the lowest prevalence was found in blood donors (0.5%). This indicates a heterogeneous distribution of Chagas disease [32].

Diagnosis of Chagas disease is difficult, as many cases are not detected by physicians until they reach the chronic stage. However, this is not the only problem. Screening blood donors for infections allows to determine the distribution of an infectious agent in the general population, as well as allowing the detection of probable cases of the disease among people who go to blood banks in an apparently adequate health condition. Although screening of blood donors can be a useful tool for early detection of Chagas disease, administrative costs for patients and physicians treating positive blood donors is a problem that delays timely care and treatment of donors who become patients. In this sense, bureaucracy plays an important role in the treatment of positive patients detected in blood banks. As soon as they are discovered, asymptomatic patients should seek medical attention. However, medications are not prescribed immediately, due to the fact that blood banks are not certified to diagnose and treat Chagas disease, even when they can perform the tests. In fact, there are alternatives besides CIA for the detection of Chagas cases that can be useful [33]. Therefore, seroreactive samples are screened with CIA, and confirmed with HAI in the blood bank. Subsequently, positive Chagas blood donors are referred as patients to the medical care unit, which, according to the recommendations of the PAHO-WHO [34], must request a third blood test from a government laboratory (in some cases, even a fourth test), and, if it is repeatedly positive, they receive the medication. Because of the administrative burden, many patients prefer not to undergo this procedure. In many cases, patients live far from medical centers that can treat this disease. Distance and lack of resources are other factors that complicate follow-up care. Public health policies related to Chagas disease should be revised to certify more laboratories and facilitate patients’ access to medication before they leave the administrative process or become chronic carriers.

Finally, brucellosis is an endemic zoonosis that can occur in different regions of the world, especially where people are exposed to increased risk factors, such as consumption of dairy products, meat from infected animals or activities related to livestock [35]. A few cases of brucellosis associated with transfusion transmission have been recorded worldwide [36,37]. For this reason, brucellosis screening has been carried out in endemic areas in Mexico for several decades and, to our knowledge, no case of brucellosis transmission by transfusion has been reported, to date.

Currently, there are no statistics that represent the epidemiological panorama of this zoonosis in our country. However, the prevalence rate of brucellosis in blood donors in a central area of Mexico is around 0.1% [12]. Records show that there were 1381 new cases of brucellosis in 2023 [26], although many people in Mexico consume unpasteurized dairy products. Remarkably, our analysis shows a sex-dependent association (*p* < 0.01, Table 2), with a similar number of positive cases in male and female donors, although the ratio of male to female donors is 2:1. This could be related to the consumption of unpasteurized dairy products rather than sex per se, although there is clinical and epidemiological evidence suggesting sexual dimorphism, including hormonal and chromosomal control influencing the prevalence of several infectious diseases such as tuberculosis or *Tropheryma whipplei*, in addition to lifestyle factors and occupational activities [38]. Sex differences in the prevalence of brucellosis are influenced by factors such as occupational exposure, cultural practices, and health-seeking behavior. In Mexico, men, who often work in agriculture, livestock farming and veterinary medicine, have more contact with animals and animal products, which increases their risk of exposure to *Brucella* bacteria. Conversely, women are more likely to go to the doctor and follow preventative measures, which reduces their risk of infection. The Central Blood Bank accepts donors from areas where livestock farming is a subsistence activity, which restricts the consumption of dairy products and increases exposure to bacteria. These areas have a low population density and are close to cities, so patients with diseases requiring frequent transfusions turn to people who donate multiple times, which could influence the presence of a higher prevalence of *Brucella* in multiple donors. Future studies must attempt to confirm and clarify this association.

Regarding *Brucella*-positive donors who became patients, the problem begins after serologic diagnosis in blood banks. Several patients receive inadequate treatment (co-trimoxazole) for seven days, and follow-up is incomplete. This is attributable to lack of knowledge and poor diffusion of the regulatory framework regarding the establishment of the appropriate treatment for brucellosis [39]. In cases when physicians have the knowledge to prescribe the correct treatment, the options are not available. Again, federal regulations should be reviewed to ensure that first-line physicians are up to date and treatment is available.

Improving public health policies for safe blood donation is crucial, but socioeconomic and structural barriers pose a major challenge [40]. Economic barriers include the high cost of testing, treatment and healthcare services, which can be prohibitively expensive, particularly for low-income communities and people with precarious employment or without insurance coverage, leading to delays or avoidance of follow-up testing. Social and cultural barriers, such as the stigma associated with sexually transmitted infections like syphilis, can prevent people from seeking follow-up care. Lack of awareness about the importance of follow-up care and the health consequences of untreated infections leads to medical advice not being followed. Cultural beliefs and practices can also influence behavior when seeking medical care. In some cultures, medical treatment for STIs is taboo, leading to a reluctance to seek follow-up care. Language barriers for non-native speakers can make it difficult to communicate with healthcare providers, making it difficult to understand the importance of follow-up care and adhere to treatment plans.

Limitations in the healthcare system and political issues also pose a challenge. Inadequate infrastructure and limited resources can hinder timely follow-up care due to a lack of medical staff, diagnostic equipment and treatment materials. Inconsistent guidelines and regulations for screening and follow-up of blood donations can create gaps in the healthcare system and lead to missed opportunities for early detection and treatment.

A multi-pronged approach is needed to remove these barriers. This includes improving access to healthcare, reducing stigma, raising awareness and implementing supportive policies. By addressing these challenges, we can improve the safety of blood donations and protect public health in Mexico and the surrounding region.

In our study, we encountered a limitation related to the follow-up of reactive donors. Not all reactive donors participated in the collection of a second specimen to confirm their results. In particular, some donors refused to attend their appointments, making it impossible to collect the required second sample. There were also cases where donors had provided incorrect personal information, making it challenging to locate them for follow-up testing. For example, Figure 1 shows that 2377 blood donors were reactive at the time of donation when they were first screened for syphilis. Of these donors, only 713 had completed their follow-up testing (35 negative on second screening; 198 negative, 2 indeterminate and 478 positive results on confirmatory testing), a follow-up rate of 31.6 percent. The rates for the other bloodborne pathogens are similar (28.7% for Chagas disease and 40.1% for brucellosis). Taking into account these follow-up rates and the total number of reactive donors at first screening, the prevalence of these three bloodborne diseases would be estimated to increase to values of around 956 cases (698 for Syphilis, 126 for Chagas disease and 132 for brucellosis) per 100,000 donors, suggesting that the problem is greater and that the prevalence of these diseases in the general population can be considered a public health problem. For this reason, it is imperative to modify the regulatory framework in Mexico to ensure the immediate confirmation of all reactive donors from the first sample and to achieve more reliable epidemiological surveillance of bloodborne diseases by performing the confirmation immediately after the first positive result, instead of following up the donor later, thus also reducing the cost and time required for the confirmatory analysis. In this context, an update of the legal framework is expected shortly, which will include positive changes to the monitoring of reactive donors. In addition, it is necessary to expand education about blood donation, including knowledge about the timely detection of bloodborne pathogens in blood donors.

## 5. Conclusions

In conclusion, bloodborne diseases are an important public health issue in Mexico and other countries. The prevalence rates reported in this study may be just the tip of the iceberg. These health problems are not unique to Mexico, as brucellosis and syphilis are medical problems worldwide. As for Chagas disease, public health policies should be reviewed to address this infection. It is necessary for federal and local authorities to address the problem and adopt public policies that focus on control and prevention, to reduce the incidence and the high costs of the long-term consequences of chronicity.

## Figures and Tables

**Figure 1 pathogens-13-01027-f001:**
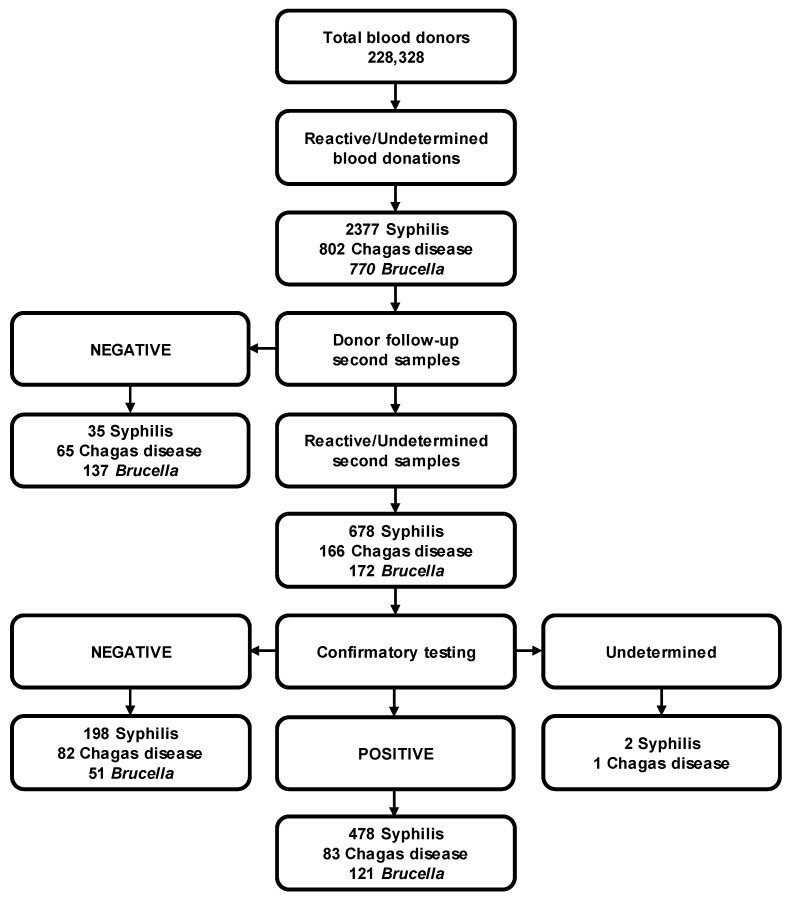
Overview of screening, follow-up and confirmation of Syphilis, Chagas disease and brucellosis in blood donors.

**Figure 2 pathogens-13-01027-f002:**
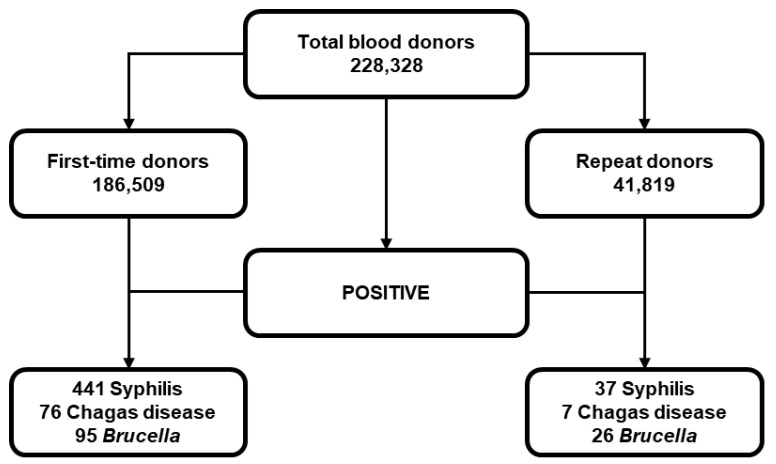
Classification of positive cases of Syphilis, Chagas disease and brucellosis, according to blood donor classification.

**Table 1 pathogens-13-01027-t001:** Distribution of first-time, repeat and total donors by sex. SD = standard deviation; CI = confidence interval.

Sex	First-Time Donors		Repeat Donors		Total	
	Number of Donors (%)	Age Means (SD, CI 95%)	Number of Donors (%)	Age Means (SD, CI 95%)	Number of Donors (%)	Age Means (SD, CI 95%)
Men	117,320 (62.90)	33.85 (10.72, 33.79–33.91)	29,546 (70.65)	33.24 (10.08, 33.12–33.35)	146,866 (64.32)	33.93 (10.60, 33.88–33.98)
Female	69,189 (37.10)	33.60 (10.82, 33.52–33.68)	12,273 (29.35)	34.17 (10.47, 33.98–34.35)	81,462 (35.68)	33.69 (10.77, 33.62–33.76)
Total	186,509 (100)	33.76 (10.76, 33.71–33.81)	41,819 (100)	34.22 (10.20, 34.12–34.32)	228,328 (100)	33.84 (10.66, 33.80–33.88)

**Table 2 pathogens-13-01027-t002:** Confirmed positive cases of non-viral bloodborne-disease blood donors and prevalence classified by sex. SD = standard deviation; CI = confidence interval; Chi square.

Non-Viral Bloodborne Disease	Sex	Positive Cases (%)	Age Means (SD, CI 95%)	Prevalence (Cases/100,000)	CI 95%	Prevalence by Sex (Cases/100,000)	CI 95%	*p*
Syphilis	Total	478 (100)	36.34 (11.01, 35.35–37.33)	209	191–229	209	191–229	0.4731
	Men	315 (65.90)	37.01 (11.38, 35.75–38.27)	138	124–154	214	192–240	
	Female	163 (34.10)	35.05 (10.18, 33.48–36.62)	71	61–83	200	172–233	
Chagas	Total	83 (100)	38.53 (10.31, 36.28–40.78)	36	29–45	36	29–45	0.7449
	Men	52 (62.65)	39.27 (10.15, 36.44–42.10)	23	17–30	35	27–46	
	Female	31 (37.35)	37.29 (10.50, 33.44–41.14)	14	10–19	38	27–54	
Brucellosis	Total	121 (100)	35.36 (9.88, 33.58–37.14)	53	44–64	53	44–64	0.0033
	Men	62 (51.24)	34.63 (10.44, 31.98–37.28)	27	21–35	42	33–54	
	Female	59 (48.76)	36.12 (9.28, 33.70–38.54)	26	20–33	72	56–93	

**Table 3 pathogens-13-01027-t003:** Confirmed positive cases and prevalences of first-time and repeat donors. SD = standard deviation; CI = confidence interval; Chi square.

First-Time Donors							
Non-Viral Bloodborne Disease	Sex	Positive Cases (%)	Age Means (SD, CI 95%)	Prevalence (Cases/100,000)	IC 95%	Prevalence BY Sex (Cases/100,000)	IC 95%
Syphilis	Men	292 (66.21)	37.12 (11.49, 35.80–38.44)	157	1.40–1.76	249	222–280
	Female	149 (33.79)	34.91 (10.30, 33.24–36.58)	80	0.68–0.94	215	184–253
	Total	441 (100)	36.38 (11.14, 35.34–37.42)	236	2.15–2.60	236	215–260
Chagas	Men	49 (64.47)	39.63 (10.20, 36.70–42.56)	26	0.20–0.35	42	32–55
	Female	27 (35.53)	36.81 (9.19, 33.17–40.44)	14	0.10–0.21	39	27–57
	Total	76 (100)	38.63 (9.88, 36.37–40.89)	41	0.32–0.51	41	32–51
Brucellosis	Men	49 (51.58)	33.76 (9.75, 30.96–36.56)	26	0.20–0.35	42	32–55
	Female	46 (48.42)	36.46 (9.85, 33.53–39.38)	25	0.18–0.33	66	50–89
	Total	95 (100)	35.06 (9.84, 33.06–37.06)	51	0.42–0.62	51	42–62
Repeat Donors							
Non-Viral Bloodborne Disease	Sex	Positive Cases (%)	Age Means (SD, CI 95%)	Prevalence (Cases/1000)	IC 95%	Prevalence by Sex (Cases/1000)	IC 95%
Syphilis	Men	23 (62.16)	35.57 (10.04, 31.23–39.91)	55	0.37–0.82	78	0.52–1.17
	Female	14 (37.84)	36.50 (8.97, 31.32–41.68)	33	0.20–0.56	114	0.68–1.91
	Total	37 (100)	35.92 (9.53, 32.74–39.10)	88	0.64–1.22	88	0.64–1.22
Chagas	Men	3 (42.86)	33.33 (8.50, 12.21–54.44)	07	0.02–0.21	10	0.03–0.30
	Female	4 (57.14)	40.50 (18.84, 10.52–70.48)	10	0.04–0.24	33	0.13–0.84
	Total	7 (100)	37.43 (14.71, 2.82–51.03)	17	0.08–0.34	17	0.08–0.34
Brucellosis	Men	13 (50.00)	37.92 (12.61, 30.30–45.54)	31	0.18–0.53	44	0.26–0.75
	Female	13 (50.00)	34.92 (7.15, 30.60–39.24)	31	0.18–0.53	106	0.62–1.81
	Total	26 (100)	36.42 (10.16, 32.32–40.52)	62	0.42–0.91	62	0.42–0.91

## Data Availability

The data supporting this study’s findings are available at the Instituto Mexicano del Seguro Social. Restrictions apply to the availability of these data, which were used under license for this study. Data are available from Daniel Ortuño-Sahagún and Erick Sierra Díaz with the permission of the Instituto Mexicano de Seguro Social.

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
