# Peer review of "Prevalence of Non-Viral Bloodborne Pathogens Among Healthy Blood Donors in Western Mexico: Problems and Failures of Public Health Policy"

_pathogens, 2024, doi:10.3390/pathogens13121027_

Round 1

Reviewer 1 Report

Comments and Suggestions for Authors

-The article addresses a problem of interest in Mexico. 

-My expertise is Chagas disease. At respect:

*In Abstract: "...but some reports indicate it also occurs in UD and Europa". I suggest change this sentence. There is a lot evidence about it. 

*The sample size and study period is important.

*Laboratory diagnosis: a): Screening: Chagas Assays kit by Chemiluminesence immunoassay. This is in accordance whith what is recommended by PHAO-WHO (See attached). b) Confirmation: the authors applied one technique (HAI Wiener Lab). For serological confirmation, is recommended the combination of two serological tests and, inclusive, a third test whne the results are inconclusive. It would be advisable, for the authors to discuss why they only used one confirmation test. 

- Discussion: it would be desirable to include information on cases of donors with confirmed Chagas disease from other LA countries who do not access medical consultation for treatment. This is a serious point. 

-What benefit does it have for the patient to diagnose teh infection if it is not going to be treated, considering the inclusion criteria regulation? 

-I think that the alerts of improved regulations in the country of origen of the study, in light of the results obtained, is the greatest value of the article. 

Author Response

Reviewer #1 Comments

  • *In Abstract: "...but some reports indicate it also occurs in UD and Europa". I suggest change this sentence. There is a lot evidence about it.

RESPONSE. We would like to thank the recommendation of the reviewer. The sentence has been changed accordingly as follows:

“Chagas disease is an endemic problem in Latin America, the United States and Europe.”

  • *Laboratory diagnosis: a): Screening: Chagas Assays kit by Chemiluminesence immunoassay. This is in accordance whith what is recommended by PHAO-WHO (See attached). b) Confirmation: the authors applied one technique (HAI Wiener Lab). For serological confirmation, is recommended the combination of two serological tests and, inclusive, a third test whne the results are inconclusive. It would be advisable, for the authors to discuss why they only used one confirmation test.

RESPONSE: We would like to thank the reviewer for his interest in this part of the study. This comment is correct, and the relevant paragraph has been expanded to clarify it as follows: (lines 324-328).

“Therefore, seroreactive samples are screened with CIA and confirmed with HAI in the blood bank. Subsequently, positive Chagas blood donors are referred as patients to medical care, which, according to PAHO-WHO recommendations [34], must request a third blood test from a government laboratory (in some cases even a fourth test), and if this is repeatedly positive, they receive the medication”.

3 - Discussion: it would be desirable to include information on cases of donors with confirmed Chagas disease from other LA countries who do not access medical consultation for treatment. This is a serious point.

RESPONSE: We would like to thank the reviewer for this comment. The data from Brazil and Colombia has been added in the Discussion section, lines 311-318.

“The prevalence of Chagas disease in blood donors has also been reported in other Latin American countries. In Pará, Brazil, the reactive rate was 0.1%, but positive cases decreased to 0.07% [31]. In Colombia, a meta-analysis showed that the prevalence of Chagas disease varied according to clinical subgroups, with a pooled prevalence of 2.0%. Higher prevalences were observed in adults (3.0%) and pregnant women (3.0%), while the lowest prevalence was found in blood donors (0.5%). This indicates a heterogeneous distribution of Chagas disease [32]”.

4 -What benefit does it have for the patient to diagnose teh infection if it is not going to be treated, considering the inclusion criteria regulation?

RESPONSE: In agreement with the reviewer's second question, we clarify this point in lines 330-335. Unfortunately, blood banks in Mexico are not certified laboratories for the diagnosis of Chagas disease. Positive blood donors must be referred to their clinical department for treatment, and the solitude for the third or fourth confirmatory test must be requested from a government laboratory. This situation is highlighted in our discussion to show the possibilities of the current health policy, as the reviewer notes in his last comment.

In agreement with the reviewer's second question, we clarify this point in lines 330-335. Unfortunately, blood banks in Mexico are not certified laboratories for the diagnosis of Chagas disease. Positive blood donors must be referred to their clinical department for treatment, and the requirement for the third or fourth confirmatory test must be performed by a government laboratory. This situation is highlighted in our discussion with the intention of showing the areas of opportunity in current health policy, as the reviewer emphasized in his last comment.

Therefore, seroreactive samples are screened with CIA, and confirmed with HAI in the blood bank. Subsequently, positive Chagas blood donors are referred as patients to medical care unit, which according to recommendations of PAHO-WHO [34] must request a third blood test from a government laboratory (in some cases even a fourth test), and if it is repeatedly positive, they receive the medication.

5 -I think that the alerts of improved regulations in the country of origen of the study, in light of the results obtained, is the greatest value of the article.

Response: We would like to thank the reviewer for the positive comments and for the careful revision of our manuscript as well as for the corrections and suggestions for improvement.

Reviewer 2 Report

Comments and Suggestions for Authors

The manuscript presented for review titled “Prevalence of non-viral bloodborne pathogens among healthy blood donors in Western Mexico: problems and failures of public health policy" highlights an often-overlooked aspect of transfusion safety by focusing on non-viral pathogens. While bloodborne disease research primarily addresses viral pathogens, this study offers valuable insights particularly relevant for low- and middle-income areas, where endemic diseases like Chagas and brucellosis continue to pose significant public health risks.

The subject matter of this article is important as it explores the prevalence of non-viral bloodborne pathogens and emphasizes measures which will improve blood safety.

The systematic analysis of the topics raised by the Authors has been presented in a clear and coherent manner. The language of the work is understandable and easy to read. The manuscript is generally well written and clear.

However, this manuscript can be improved

Introduction

The authors should expand the Introduction with section where they would introduce the manuscript’s goal by explaining why these specific non-viral pathogens have been designated as mandatory for testing and how they affect public health in Mexico and the surrounding region

Discussion

The discussion emphasizes the importance of enhancing public health policies. Authors should provide more data with the international comparisons. Providing a more in-depth analysis of socioeconomic barriers (potential costs) to follow-up would offer additional context to the finding.

The authors should expand the discussion by analyzing additional risk factors contributing to the significant gender difference in brucellosis prevalence, beyond the consumption of unpasteurized dairy. They should also provide their analysis of why it occurs more frequently among repeat donors.

Provide a comparative analysis of infection rates and behaviors between first-time and repeat donors and also compare it to other researches in Mexico or surrounding countries.

Author Response

Reviewer #1 Comments

General comments:

The manuscript presented for review titled “Prevalence of non-viral bloodborne pathogens among healthy blood donors in Western Mexico: problems and failures of public health policy" highlights an often-overlooked aspect of transfusion safety by focusing on non-viral pathogens. While bloodborne disease research primarily addresses viral pathogens, this study offers valuable insights particularly relevant for low- and middle-income areas, where endemic diseases like Chagas and brucellosis continue to pose significant public health risks.

The subject matter of this article is important as it explores the prevalence of non-viral bloodborne pathogens and emphasizes measures which will improve blood safety.

The systematic analysis of the topics raised by the Authors has been presented in a clear and coherent manner. The language of the work is understandable and easy to read. The manuscript is generally well written and clear.

However, this manuscript can be improved

ANSWER: We would like to thank the reviewer for his positive comments on our work and for the time taken to review our manuscript in such detail. Below you will find the answers to the reviewer's queries.

Details comments:

  • Introduction. The authors should expand the Introduction with section where they would introduce the manuscript’s goal by explaining why these specific non-viral pathogens have been designated as mandatory for testing and how they affect public health in Mexico and the surrounding region

ANSWER: We thank the reviewer for the opportunity to go deep into this aspect. In agreement with this, we have included the requested information (lines 57-65)

In Mexico, syphilis, Chagas disease and brucellosis pose serious transmission risks from contaminated blood transfusions, which have a negative impact on public health and lead to the spread of disease and severe and chronic complications in the recipient population. In addition, Mexico is considered an endemic country for Chagas disease due to its climate and the presence of triatomines [5], and for brucellosis due to the high consumption of unpasteurized dairy products. To ensure the safety of the blood supply and protect public health from these potentially serious infections, it is therefore essential to test blood donations for these diseases. This proactive approach helps to reduce the incidence of transfusion-transmitted infections and the associated health burden.

  • The discussion emphasizes the importance of enhancing public health policies. Authors should provide more data with the international comparisons. Providing a more in-depth analysis of socioeconomic barriers (potential costs) to follow-up would offer additional context to the finding.

ANSWER: We thank the reviewer for pointing out this relevant aspect. We have added the information requested as follows (lines 370-391):

Improving public health policies for safe blood donation is crucial, but socioeconomic and structural barriers pose a major challenge [37]. Economic barriers include the high cost of testing, treatment and healthcare services, which can be prohibitively expensive, particularly for low-income communities and people with precarious employment or without insurance coverage, leading to delays or avoidance of follow-up testing. Social and cultural barriers, such as the stigma associated with sexually transmitted infections like syphilis, can prevent people from seeking follow-up care. Lack of awareness about the importance of follow-up care and the health consequences of untreated infections leads to medical advice not being followed. Cultural beliefs and practices can also influence behavior when seeking medical care. In some cultures, medical treatment for STIs is taboo, leading to a reluctance to seek follow-up care. Language barriers for non-native speakers can make it difficult to communicate with healthcare providers, making it difficult to understand the importance of follow-up care and adhere to treatment plans.

Limitations in the healthcare system and political issues also pose a challenge. Inadequate infrastructure and limited resources can hinder timely follow-up care due to a lack of medical staff, diagnostic equipment and treatment materials. Inconsistent guidelines and regulations for screening and follow-up of blood donations can create gaps in the healthcare system and lead to missed opportunities for early detection and treatment.

A multi-pronged approach is needed to remove these barriers. This includes improving access to healthcare, reducing stigma, raising awareness and implementing supportive policies. By addressing these challenges, we can improve the safety of blood donations and protect public health in Mexico and the surrounding region.

Martin K, Wenlock R, Roper T, Butler C, Vera JH. Facilitators and barriers to point-of-care testing for sexually transmitted infections in low- and middle-income countries: a scoping review.

BMC Infect Dis. 2022 Jun 20;22(1):561. doi: 10.1186/s12879-022-07534-9.

  • The authors should expand the discussion by analyzing additional risk factors contributing to the significant gender difference in brucellosis prevalence, beyond the consumption of unpasteurized dairy. They should also provide their analysis of why it occurs more frequently among repeat donors.

ANSWER: We thank the reviewer for the opportunity to address this aspect in deep. Consequently, we have added information as follows (lines 350-360):

In Mexico, men, who often work in agriculture, livestock farming and veterinary medicine, have more contact with animals and animal products, which increases their risk of exposure to Brucella bacteria. Conversely, women are more likely to go to the doctor and follow preventative measures, which reduces their risk of infection. The Central Blood Bank accepts donors from areas where livestock farming is a subsistence activity, which restricts the consumption of dairy products and increases exposure to bacteria. These areas have a low population density and are close to cities, so patients with diseases requiring frequent transfusions turn to people who donate multiple times, which could influence the presence of a higher prevalence of Brucella in multiple donors.

  • Provide a comparative analysis of infection rates and behaviors between first-time and repeat donors and also compare it to other researches in Mexico or surrounding countries.

ANSWER: We thank the reviewer for the opportunity to elaborate on this aspect. We have included the following more detailed information in response to this aspect:

Lines 220-221: The prevalence of syphilis among blood donors in Mexico varies, with some studies re-porting rates of around 2.1% [12].

Lines 339-340: However, the prevalence rate of brucellosis in blood donors in a central area of Mexico is around 0.1% [12].

Lines 306-308: In blood donors, the prevalence of Chagas disease is estimated to be around 0.65% [12], but there are no studies that allow us to get an idea of the epidemiological situation in Mexico.

Med Int Méx. 2020 enero-febrero;36(1):15-20. Prevalencia de serología de enfermedades infecciosas en donadores de sangre durante 17 años en Guanajuato, México. Prevalence of serology for infectious diseases in blood donors during 17 years in Guanajuato, Mexico. Marco V Sangrador-Deitos,1 Álvaro Cruz-Hernández,1 Jimena A González-Olvera,1 Luis Alberto Rodríguez-Hernández,1 Carlos Daniel Sánchez-Cárdenas,1 Fernando G Torres-Salgado2

According to the study by Osei-Boakye et.al. (2024), first-time donors are more likely to report high-risk behaviors such as multiple sexual partners, drug use, and recent tattoos or piercings. In addition, first-time donors have a higher deferral rate due to these behaviors and other factors such as low hemoglobin levels. Repeat donors, on the other hand, generally report less risky behaviors and are better informed about safe practices. In addition, repeat donors generally have a healthier profile, fewer refusals and better overall health indicators [23].

Self-reported high-risk behavior among first-time and repeat replacement blood donors; a four-year retrospective study of patterns.

Osei-Boakye F, Nkansah C, Appiah SK, Abbam G, Derigubah CA, Ukwah BN, Usanga VU, Ugwuja EI, Chukwurah EF. PLoS One. 2024 Aug 8;19(8):e0308453. doi: 10.1371/journal.pone.0308453.

Reviewer 3 Report

Comments and Suggestions for Authors

This is an interesting study aimed to determine the prevalence on 3 non-viral bloodborne diseases among blood donors in Mexico. Some comments and suggestions are presented below, and need to be addressed by the Authors.

Positive results for viral diseases were not considered for this study (line 84). This is somewhat concerning, because due to common risk factors, co-infection with HIV is not uncommon; moreover syphilis shows atypical characteristics in the HIV-positive patient and favour the viral spread and the progression of the disease. If possible, add data on syphilis-HIV coinfection

Line 99. could you give some information on  what NOM-253-SSA1-2012 is and references?

M&M, lines 94-7 “If a sample was reactive or undetermined for Treponema pallidum or Trypanosoma cruzi, a sample of the plasma was collected as prescribed by the manufacturer, and the analysis was repeated for both samples simultaneously.” However, only 1253 positive donors (34.4% of the total positive individuals) were located, and their follow-up completed (lines 140-1). This is an important limitation of the validity of the study findings, and need to be stated starting from the abstract, in results and discussion. Also add in fig. 1 the number of positive tests at first screening without confirmation test. Of those who underwent confirmatory tests,  331 (around ¼) were negative. 

BIO-RAD TPHA was used as confirmatory test for syphilis, and donors with positive confirmatory tests were considered positive cases for infection and their respective disease (lines 107-8). How were the confirmed positive cases informed  and  counselled? Positivity for treponemal tests encompass cases of active diseases (primary, secondary) latent cases, and also previous cases treated successfully. You have to be more clear on this important point. Were the positive cases sent to a specialist for counselling and treatment when required?

Non-venereal treponematoses (e.g., yaws, endemic syphilis, and pinta) constitute a major health concern for many  countries, including Mexico. These diseases are caused  organisms that are morphologically and antigenically identical to the causative agent of venereal syphilis, Treponema pallidum. Could be these non-venereal treponematosis responsible of false positive results for the screening and confirmatory tests for syphilis?

               Are you able to provide more information as to what the risk factors for syphilis and T. cruzi were? Also, are you also able to compare the rates of infection in your population of donors vs the general population in your country?

Unexpectedly, compared to data from other reports and in the general population, prevalence of T. pallidum and T. cruzi were similar among men and females, and  this was consistent in the population study, in first time donors and repeat donors. A consistent increase of cases of syphilis has been observed worldwide among MSM, but it seems that this  had no impact on the current population of Mexican blood donors. A possibility for this unexpected trend is related, as stated in lines 216-8, to the exclusion of donors  due to risk factors related to sexual practices, as required by blood banks regulations in Mexico. Another possibility is an unbalance between the completed follow-up and availability of confirmation tests between men and women (for example more women undergoing  confirmatory tests compared to men, including MSM).  Please expand this aspect of the data, and add more information on the population who completed the follow-up.

Author Response

Reviewer #2 Comments

This is an interesting study aimed to determine the prevalence on 3 non-viral bloodborne diseases among blood donors in Mexico. Some comments and suggestions are presented below, and need to be addressed by the Authors.

We would like to thank the reviewer for his positive comments on our work and for the time taken to review our manuscript. Below you will find the answers to the reviewer's suggestions.

  • Positive results for viral diseases were not considered for this study (line 84). This is somewhat concerning, because due to common risk factors, co-infection with HIV is not uncommon; moreover syphilis shows atypical characteristics in the HIV-positive patient and favour the viral spread and the progression of the disease. If possible, add data on syphilis-HIV coinfection

We agree with the reviewer's comment that the presence of HIV contributes as a risk factor for the spread of syphilis. However, the aim of our study is to highlight an often-overlooked aspect of transfusion safety by focusing on non-viral pathogens. In lines 46-50, a sentence has been added and reference made to our previous study on HIV, HCV and HBV seroprevalences.

  • Line 99. could you give some information on what NOM-253-SSA1-2012 is and references?

ANSWER: NOM-253-SSA1-2012 has been defined and its purpose explained in the introduction. The reference has also been included (lines 65-74).

In this sense, federal regulations (NOM-253-SSA1_2012) stipulate that all blood donors must be tested for HIV, HBV, HCV, syphilis, Trypanosoma cruzi (Chagas disease) and - depending on the region and epidemiology - for other blood-borne pathogens: Brucella, Plasmodium, Cytomegalovirus, Toxoplasma and Human T-lymphotropic virus (HTLV) 1 and 2. The NOM-253-SSA1-2012 is the national framework that governs the blood supply and emphasizes the importance of encouraging voluntary, unpaid blood donations to ensure a safe and sufficient supply. It establishes strict standards for the collection, testing, processing and distribution of blood and its components to minimize transfusion-transmitted infections [6].

  • M&M, lines 94-7 “If a sample was reactive or undetermined for Treponema pallidum or Trypanosoma cruzi, a sample of the plasma was collected as prescribed by the manufacturer, and the analysis was repeated for both samples simultaneously.” However, only 1253 positive donors (34.4% of the total positive individuals) were located, and their follow-up completed (lines 140-1). This is an important limitation of the validity of the study findings, and need to be stated starting from the abstract, in results and discussion. Also add in fig. 1 the number of positive tests at first screening without confirmation test. Of those who underwent confirmatory tests, 331 (around ¼) were negative.

ANSWER: A sentence was added in the abstract to complement it and alineate with the results section and discussion. The number of positive test are named as “reactive/undetermined” in the figure 1. Those are the terms used by the federal regulations, and correspond to first screening and donor follow-up with the second samples. Also, negative samples are included at both, first screening and follow-up. “Reactive and undetermined” samples are defined in material and methods.

  • BIO-RAD TPHA was used as confirmatory test for syphilis, and donors with positive confirmatory tests were considered positive cases for infection and their respective disease (lines 107-8). How were the confirmed positive cases informed and counselled? Positivity for treponemal tests encompass cases of active diseases (primary, secondary) latent cases, and also previous cases treated successfully. You have to be more clear on this important point. Were the positive cases sent to a specialist for counselling and treatment when required?

ANSWER: We would like to thank the reviewer for his comment. A sentence has been added to the same paragraph to clarify the process (lines 128-129). Positive cases are referred to their medical care unit for clinical care and treatment.

  • Non-venereal treponematoses (e.g., yaws, endemic syphilis, and pinta) constitute a major health concern for many countries, including Mexico. These diseases are caused organisms that are morphologically and antigenically identical to the causative agent of venereal syphilis, Treponema pallidum. Could be these non-venereal treponematosis responsible of false positive results for the screening and confirmatory tests for syphilis?

ANSWER: We would like to thank to reviewer for their comment. A sentence was added in the

discussion (lines 226-230) as follows:

However, it is important to keep in mind that the detection of anti-T. pallidum antibodies by CIA has a higher false-positive rate in populations with low syphilis prevalence [15]. In addition, it is worth noting that non-venereal treponematoses such as soft palate, endemic syphilis and pinta can cause false-positive results in syphilis screening and confirmation tests [16].

https://www.intechopen.com/chapters/83264     DOI: 10.5772/intechopen.106370

  • Are you able to provide more information as to what the risk factors for syphilis and T. cruzi were? Also, are you also able to compare the rates of infection in your population of donors vs the general population in your country?

ANSWER: We appreciate the reviewer's interest in this part of the study. In accordance with regulatory requirements, all donors will be interviewed and those who are eligible will not declare risk factors for these diseases. If they do, they must be deferred. Therefore, the positive cases that are reported are usually people who are hiding information. As for your second approach, the available information is controversial. Unfortunately, the few studies that exist on this subject are sparse and have been conducted in small populations, as they concern blood banks with fewer donations than ours. This also highlights the relevance of our study.

  • Unexpectedly, compared to data from other reports and in the general population, prevalence of T. pallidum and T. cruzi were similar among men and females, and this was consistent in the population study, in first time donors and repeat donors. A consistent increase of cases of syphilis has been observed worldwide among MSM, but it seems that this had no impact on the current population of Mexican blood donors. A possibility for this unexpected trend is related, as stated in lines 216-8, to the exclusion of donors due to risk factors related to sexual practices, as required by blood banks regulations in Mexico. Another possibility is an unbalance between the completed follow-up and availability of confirmation tests between men and women (for example more women undergoing confirmatory tests compared to men, including MSM). Please expand this aspect of the data, and add more information on the population who completed the follow-up.

ANSWER: We would like to thank you for your comment. The analysis did not show that the prevalence of syphilis is statistically dependent on gender. Therefore, the paragraph has been amended as follows (lines 242-249):

The regulations for blood banks in Mexico take into account the exclusion of donors based on risk factors related to sexual practices, such as contact with secretions and body fluids of persons who may have sexually transmitted diseases. However, these results are underestimated by the existence of a bias, as not all reactive/unsafe blood donors complete their follow-up and positive cases cannot be confirmed, as we explain later in the limitations of the study. However, analysis of the data suggests that syphilis prevalence is not dependent on donor gender. It is therefore possible that donors who did not complete their follow-up may maintain this statistical trend.

Round 2

Reviewer 2 Report

Comments and Suggestions for Authors

The authors have revised the manuscript and accepted and corrected all the given suggestions. In this form, the revised manuscript offers a significant contribution to the field of non-viral bloodborne pathogens, their prevalence and significance in blood transfusion safety. This research can provide a foundation for further studies and potentially contribute to changes in public policies which can help prevent diseases and promote health.
I am pleased to recommend that the manuscript should be published in the Journal without any additional revisions.

Author Response

We thank the reviewer for his positive opinion. Since another reviewer commented on our manuscript after we had replied to the first two reviewers, we include here the responses to this late revision:

Reviewer #1 Late comments

  • *In Abstract: "...but some reports indicate it also occurs in UD and Europa". I suggest change this sentence. There is a lot evidence about it.

RESPONSE. We would like to thank the recommendation of the reviewer. The sentence has been changed accordingly as follows:

“Chagas disease is an endemic problem in Latin America, the United States and Europe.”

  • *Laboratory diagnosis: a): Screening: Chagas Assays kit by Chemiluminesence immunoassay. This is in accordance whith what is recommended by PHAO-WHO (See attached). b) Confirmation: the authors applied one technique (HAI Wiener Lab). For serological confirmation, is recommended the combination of two serological tests and, inclusive, a third test whne the results are inconclusive. It would be advisable, for the authors to discuss why they only used one confirmation test.

RESPONSE: We would like to thank the reviewer for his interest in this part of the study. This comment is correct, and the relevant paragraph has been expanded to clarify it as follows: (lines 324-328).

“Therefore, seroreactive samples are screened with CIA and confirmed with HAI in the blood bank. Subsequently, positive Chagas blood donors are referred as patients to medical care, which, according to PAHO-WHO recommendations [34], must request a third blood test from a government laboratory (in some cases even a fourth test), and if this is repeatedly positive, they receive the medication”.

3 - Discussion: it would be desirable to include information on cases of donors with confirmed Chagas disease from other LA countries who do not access medical consultation for treatment. This is a serious point.

RESPONSE: We would like to thank the reviewer for this comment. The data from Brazil and Colombia has been added in the Discussion section, lines 311-318.

“The prevalence of Chagas disease in blood donors has also been reported in other Latin American countries. In Pará, Brazil, the reactive rate was 0.1%, but positive cases decreased to 0.07% [31]. In Colombia, a meta-analysis showed that the prevalence of Chagas disease varied according to clinical subgroups, with a pooled prevalence of 2.0%. Higher prevalences were observed in adults (3.0%) and pregnant women (3.0%), while the lowest prevalence was found in blood donors (0.5%). This indicates a heterogeneous distribution of Chagas disease [32]”.

4 -What benefit does it have for the patient to diagnose teh infection if it is not going to be treated, considering the inclusion criteria regulation?

RESPONSE: In agreement with the reviewer's second question, we clarify this point in lines 330-335. Unfortunately, blood banks in Mexico are not certified laboratories for the diagnosis of Chagas disease. Positive blood donors must be referred to their clinical department for treatment, and the solitude for the third or fourth confirmatory test must be requested from a government laboratory. This situation is highlighted in our discussion to show the possibilities of the current health policy, as the reviewer notes in his last comment.

In agreement with the reviewer's second question, we clarify this point in lines 330-335. Unfortunately, blood banks in Mexico are not certified laboratories for the diagnosis of Chagas disease. Positive blood donors must be referred to their clinical department for treatment, and the requirement for the third or fourth confirmatory test must be performed by a government laboratory. This situation is highlighted in our discussion with the intention of showing the areas of opportunity in current health policy, as the reviewer emphasized in his last comment.

Therefore, seroreactive samples are screened with CIA, and confirmed with HAI in the blood bank. Subsequently, positive Chagas blood donors are referred as patients to medical care unit, which according to recommendations of PAHO-WHO [34] must request a third blood test from a government laboratory (in some cases even a fourth test), and if it is repeatedly positive, they receive the medication.

5 -I think that the alerts of improved regulations in the country of origen of the study, in light of the results obtained, is the greatest value of the article.

Response: We would like to thank the reviewer for the positive comments and for the careful revision of our manuscript as well as for the corrections and suggestions for improvement.

Reviewer 3 Report

Comments and Suggestions for Authors

all the points raised have been addressed

Author Response

(The authors gave the same response as above.)
